

# Weather conditions associated with autumn migration by mule deer in Wyoming

Chadwick D. Rittenhouse[1], Tony W. Mong[2] and Thomas Hart[3]

[1] Department of Natural Resources and the Environment, Wildlife and Fisheries Conservation Center, University of Connecticut, Storrs, CT, USA
[2] Wyoming Game and Fish Department, Savery, WY, USA
[3] Environmental Services Section, Wyoming Department of Transportation, Cheyenne, WY, USA

## ABSTRACT

Maintaining ecological integrity necessitates a proactive approach of identifying and acquiring lands to conserve unfragmented landscapes, as well as evaluating existing mitigation strategies to increase connectivity in fragmented landscapes. The increased use of highway underpasses and overpasses to restore connectivity for wildlife species offers clear conservation benefits, yet also presents a unique opportunity to understand how weather conditions may impact movement of wildlife species. We used remote camera observations (19,480) from an existing wildlife highway underpass in Wyoming and daily meteorological observations to quantify weather conditions associated with autumn migration of mule deer in 2009 and 2010. We identified minimal daily temperature and snow depth as proximate cues associated with mule deer migration to winter range. These weather cues were consistent across does and bucks, but differed slightly by year. Additionally, extreme early season snow depth or cold temperature events appear to be associated with onset of migration. This information will assist wildlife managers and transportation officials as they plan future projects to maintain and enhance migration routes for mule deer.

# INTRODUCTION

Maintaining ecological integrity necessitates a proactive approach of identifying threats to species, communities, and the ecological processes that sustain them. For mule deer (*Odocoileus hemionus*), a culturally and economically important species (*Copeland et al., 2014*), habitat fragmentation can present barriers to migration, alter migration routes, or increase mortality through deer-vehicle collisions (*Sawyer, LeBeau & Hart, 2012*).

Highway underpasses have restored connectivity for mule deer in fragmented landscapes (*Reed, Woodward & Pojar, 1975*; *Ng et al., 2004*; *Clevenger & Waltho, 2005*; *Braden et al., 2008*; *Gagnon et al., 2011*). The unique structure of highway underpasses and associated fencing, when coupled with remote cameras and weather observations, presents an opportunity to gain substantial information on population age and sex structure

Corresponding author
Chadwick D. Rittenhouse,
chadwick.rittenhouse@uconn.edu

(*Ikeda et al., 2013*), abundance (*Rowcliffe et al., 2008*), and proximate cues associated with migratory movement. Yet, the relationships between migratory movements and weather conditions are understudied, despite the importance of this for informing policy and adaptive management decisions regarding connectivity in the context of a changing climate.

Our objective was to identify weather conditions associated with autumn migration by mule deer in Wyoming. Our central hypothesis is that entering the winter range too early comes at the expense of reproductive output and survival of young mule deer. Our basis for this hypothesis arises from known nutritional limitations and energetic costs of winter to mule deer (*Bartmann, White & Carpenter, 1992*; *Poole & Mowat, 2005*; *Bishop et al., 2009*); premature entry to the winter grounds presumably provides no advantage to adults or offspring. Based on this hypothesis, we predict that mule deer movements through the underpass will be associated with the timing of snowfall events and snow depth that cover forage on summer grounds. Identification of the specific cue(s) used by mule deer during migration may allow managers to anticipate changes in migration based on weather conditions.

## METHODS

### Study site and camera set-up

We collected animal migration data using a trail camera set up to monitor a highway underpass on Hwy. 789 approximately 8 km north of Baggs, Wyoming (Fig. 1). The underpass was installed in 2009 and equipped with a RECONYX Hyperfire (TM) camera (Reconyx, Holmen, Wisconsin, USA) mounted approximately 1.5 m high centered within the underpass. The camera was pointed towards the direction animals were migrating from, so during autumn the camera was pointed east. Camera settings included a distance from camera to subjects of 18.2 m with a 1/5 s trigger speed; three photos recorded when motion was detected, and photo resolution of 1080P High Definition or 3.1 Mega-pixels. Cameras remained active throughout the year; however, during autumn migrations by mule deer the images were downloaded more frequently. Autumn migration dates were November 1 to December 31 of each year. The Wyoming Game and Fish Department approved this research (Permit #791).

The study site was defined by the Baggs Mule Deer Herd Unit, which encompasses 1,092 km$^2$ south of I-80 in southern Carbon County (Fig. 1). This area supports a variety of vegetation types, but is generally characterized by rolling topography, prominent ridges, and dry canyons dominated by sagebrush (*Artemisia sp.*), black greasewood (*Sacrobatus vermiculatus*), Utah juniper (*Juniperus osteosperma*), and other mixed-shrub (*Purshia tridentata, Prunus virginiana, Amelanchier alnifolia, Chrysothamnus sp., Cercocarpus sp.*). Elevations range from 1,920 to 2,530 m.

### Population age and sex structure

We used images taken by the camera to count and assign age class (fawn, yearling, adult) and sex to all deer passing through the underpass during autumn migrations 2009 and 2010. Does and fawns migrated together, as did adult and yearling bucks. Therefore, we

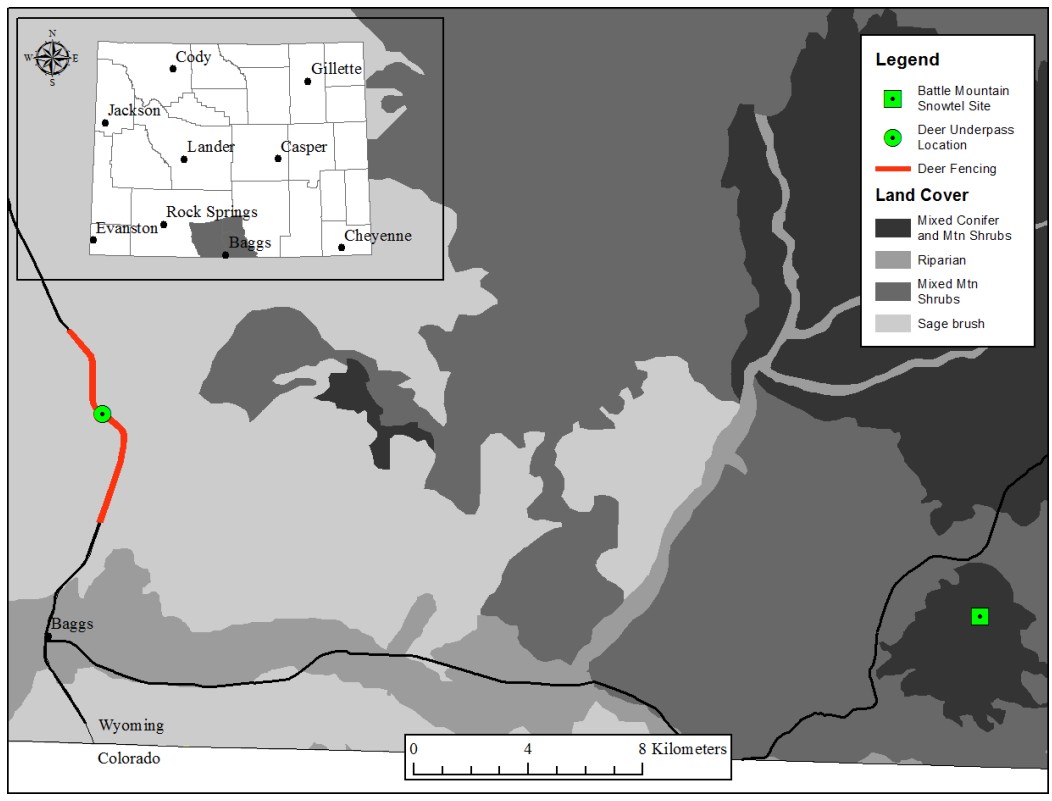

**Figure 1 Study area near Baggs, Wyoming.** Location of the highway underpass and fencing, and the meteorological station, near Baggs, Wyoming. During autumn, mule deer migrate from the higher elevation summer range north and east of the underpass to lower elevation winter range to the west of the highway that contains the underpass.

tallied does separate from fawn, yearling buck, and adult classes. While most deer moved through the underpass one way, in some instances multiple images were obtained of the same individuals due to the three-image sequence provided by the camera. When this occurred, we used group size, group composition, and size of individuals to count unique individuals only.

Common indices used by wildlife managers to monitor population sex and age structure include: ratio of adult doe per fawn, ratio of adult doe per yearling buck, and ratio of adult doe per adult buck. We quantified those ratios for fall migration of each year and provide SE and 95% CI.

## Association of migratory movements with weather conditions

We obtained meteorological records from the Battle Creek weather station (31 km from underpass, Natural Resources Conservation Service Snotel Site number 317, 41°; 3 min N, 107°; 16 min W). The meteorological records consisted of daily records of maximum air temperature (°C), minimum air temperature (°C), average air temperature (°C), precipitation accumulation (in), snow water equivalent (in), and snow depth (in). From those weather observations, we calculated one additional metric: snow event. Snow event had a value of 1 if snow fell on that day. If no snow fell on that day we assigned a value

**Table 1 Counts of mule deer by age and sex classes.** Age and sex counts and ratios of mule deer in the Baggs District of Wyoming observed from a camera trap fixed to a highway underpass during autumn migration of 2009 and 2010.

| Year | Sex-age class | N | Ratio of does per sex-age class | SE | −95% CI | +95% CI |
|------|---------------|---|--------------------------------|----|---------|---------|
| 2009 | Does | 1,205 | | | | |
| | Fawns | 754 | 1.60 | 0.07 | 1.45 | 1.74 |
| | Yearling bucks | 81 | 14.88 | 1.71 | 11.53 | 18.22 |
| | Adult bucks | 170 | 7.09 | 0.58 | 5.95 | 8.23 |
| 2010 | Does | 2,401 | | | | |
| | Fawns | 1,441 | 1.67 | 0.06 | 1.56 | 1.78 |
| | Yearling bucks | 228 | 10.53 | 0.73 | 9.10 | 11.96 |
| | Adult bucks | 315 | 7.62 | 0.46 | 6.73 | 8.52 |

of 0 to snow event. Preliminary analyses indicated high correlations ($R > 0.50$) among all weather variables, so we proceeded with models containing each weather variable separately. Preliminary analyses also indicated that weather observations from the previous day were better predictors of daily migration counts than same day observations because they accounted for the lag between a weather event at higher elevation and deer arrival at the underpass. Therefore, all weather conditions except snow depth reflected the previous day's weather.

To identify weather conditions associated with autumn migration by mule deer, we modeled counts of does and bucks (yearling and adult combined) as a function of weather variables. The counts were specified as the response variable in separate models of minimum air temperature, maximum air temperature, precipitation amount, snowfall event, and snow depth as independent variables. For each model, we included each independent weather variable and an interaction term with day of the migration season, with day 1 corresponding to 17 October 2009 and 8 October 2010. We also examined a model that consisted of all independent variables and their interactions with day of migration season.

We used negative binomial regression models to handle overdispersed count data with 0 observations. We used Akaike's information criterion (AIC) to rank models and Akaike weights ($w_i$) to determine which model associating migration with weather conditions had the strongest support (*Burnham & Anderson, 2002*). We fitted all models with the glm.nb function in the R language and environment for statistical analyses (version 2.15.2) (*R Core Development Team, 2012*).

## RESULTS

Analysis of 19,480 images acquired during fall migration 2009 and fall migration 2010 documented 6,628 counts of mule deer using the underpass (Table 1). The population age and sex structure was consistent across the two years with a mean of 61 fawns, 8 yearling bucks, and 14 adult bucks per 100 does (Table 1).

Model-selection results indicated minimum air temperature and snow depth were the best proximate cues associated with autumn migration. In 2009, minimum air temperature

**Table 2 AIC ranks for models associating weather and mule deer migration.** Ranked empirical support for models examining how weather conditions influence autumn migration by mule deer in the Baggs District of Wyoming. All models included day as a variable, but the variable name was omitted for brevity. Data collected from a camera trap fixed to a highway underpass during autumn migration of 2009 and 2010.

| Sex | Year | Model | AIC | Δ AIC | $w_i$[a] | $r^2$[b] |
|-----|------|-------|-----|-------|-------|------|
| Does | 2009 | Min air temp, previous day | 517.73 | 0.00 | 0.83 | 0.42 |
| | | Max air temp, previous day | 521.10 | 3.36 | 0.15 | 0.39 |
| | | Full model | 525.76 | 8.03 | 0.02 | 0.39 |
| | | Snow depth | 540.58 | 22.85 | 0.00 | 0.19 |
| | | Snowfall event, previous day | 548.84 | 31.11 | 0.00 | 0.08 |
| | | Precip. amount, previous day | 550.21 | 32.47 | 0.00 | 0.06 |
| | 2010 | Snow depth | 616.34 | 0.00 | 0.95 | 0.58 |
| | | Min air temp, previous day | 622.12 | 5.79 | 0.05 | 0.55 |
| | | Max air temp, previous day | 629.18 | 12.85 | 0.00 | 0.50 |
| | | Snowfall event, previous day | 647.14 | 30.81 | 0.00 | 0.37 |
| | | Precip. amount, previous day | 650.81 | 34.48 | 0.00 | 0.34 |
| | | Full model | 655.28 | 38.94 | 0.00 | 0.33 |
| Bucks | 2009 | Min air temp, previous day | 329.55 | 0.00 | 0.43 | 0.26 |
| | | Max air temp, previous day | 329.87 | 0.32 | 0.37 | 0.26 |
| | | Snow depth | 332.39 | 2.84 | 0.10 | 0.23 |
| | | Full model | 332.49 | 2.94 | 0.10 | 0.27 |
| | | Snowfall event, previous day | 344.95 | 15.39 | 0.00 | 0.07 |
| | | Precip. amount, previous day | 348.70 | 19.15 | 0.00 | 0.02 |
| | 2010 | Snow depth | 405.98 | 0.00 | 1.00 | 0.58 |
| | | Min air temp, previous day | 426.15 | 20.17 | 0.00 | 0.46 |
| | | Max air temp, previous day | 434.34 | 28.36 | 0.00 | 0.39 |
| | | Full model | 446.80 | 40.82 | 0.00 | 0.32 |
| | | Snowfall event, previous day | 448.51 | 42.53 | 0.00 | 0.27 |
| | | Precip. amount, previous day | 452.30 | 46.31 | 0.00 | 0.23 |

**Notes.**

[a] Weights of evidence.

[b] Fitted model versus null model (*Magee, 1990*).

of the previous day was the most supported model for does and bucks, with maximum air temperature of the previous day competing with the most supported model for bucks only (Table 2). In 2010, snow depth was the most supported model for does and bucks with no competing models for either sex. Higher counts of deer occurred on days with lower minimum temperatures in 2009 and on days with greater snow depth in 2010 (Table 3). Bucks and does responded similarly to weather conditions within and across years despite migrating in separate groups.

We used coefficients from the top fitted models to predict use of the highway underpass under the range of weather conditions observed in 2009 and 2010 (Fig. 2). Early, extreme minimum air temperatures and snow depths had the highest predicted counts of does and bucks. Based on these model predictions, thresholds for onset of migration were minimum air temperature of 0 to −5 °C or snow depth exceeding 25.4 cm.

**Table 3 Influence of weather on autumn migration by mule deer.** Parameter estimates, standard errors, *z* scores, and *p*-values for the most-supported models examining how weather conditions influence autumn migration by mule deer in the Baggs District of Wyoming. Data collected from a camera trap fixed to a highway underpass during autumn migration of 2009 and 2010.

| Sex | Year | Parameter | Estimate | Std. error | z value | P-value |
|---|---|---|---|---|---|---|
| Does | 2009 | Intercept | 2.217 | 0.245 | 9.065 | <0.001 |
| | | Min air temp, previous day | −0.206 | 0.038 | −5.364 | <0.001 |
| | | Day | 0.011 | 0.009 | 1.193 | 0.233 |
| | | Min air temp, previous day:Day | 0.004 | 0.001 | 3.719 | <0.001 |
| | 2010 | Intercept | 1.229 | 0.253 | 4.854 | <0.001 |
| | | Snow depth | 0.371 | 0.057 | 6.562 | <0.001 |
| | | Day | 0.052 | 0.010 | 5.426 | <0.001 |
| | | Snow depth:Day | −0.007 | 0.001 | −7.328 | <0.001 |
| Bucks | 2009 | Intercept | 1.300 | 0.272 | 4.773 | <0.001 |
| | | Min air temp, previous day | −0.127 | 0.042 | −3.044 | 0.002 |
| | | Day | −0.011 | 0.010 | −1.091 | 0.275 |
| | | Min air temp, previous day:Day | 0.002 | 0.001 | 1.409 | 0.159 |
| | 2010 | Intercept | 0.571 | 0.238 | 2.401 | 0.02 |
| | | Snow depth | 0.402 | 0.047 | 8.487 | <0.001 |
| | | Day | 0.026 | 0.009 | 2.945 | 0.003 |
| | | Snow depth:Day | −0.007 | 0.001 | −7.910 | <0.001 |

## DISCUSSION

Wildlife underpasses and overpasses are used throughout western North America to restore connectivity for migratory and large-ranging species. We demonstrated that monitoring wildlife underpasses provides important demographic and movement information, which when coupled with weather observations can be used to understand connectivity and migratory dynamics associated with weather conditions. The association between weather conditions and counts of deer using the underpass was consistent with our expectation. Specifically, we identified minimal daily temperature and snow depth as proximate cues used by mule deer during migration to winter range. These weather cues were consistent across does and bucks, but differed slightly by year. Additionally, extreme early season snow depth exceeding 25.4 cm or minimum air temperature of 0 to −5 °C may be associated with onset of migration.

Our results using a single camera trap were consistent with radio-telemetry studies of mule deer migration. Previous studies identified decreasing daily temperature, increasing snow depth, and senescing vegetation as factors associated with autumn migration from high-elevation summer ranges to low-elevation winter ranges (*Garrott et al., 1987*; *Nicholson, Bowyer & Kie, 1997*; *Monteith et al., 2011*). However, the specific relationships varied by vegetation type, elevation, and climate. In the sagebrush-steppe ecosystem of the Sierra Nevada Mountains in California, the onset of migration coincided with average daily temperature <5 °C and snow depth >0 cm (*Monteith et al., 2011*). In a pinyon pine-Utah juniper shrubland complex in northwestern Colorado, mule deer migration occurred with
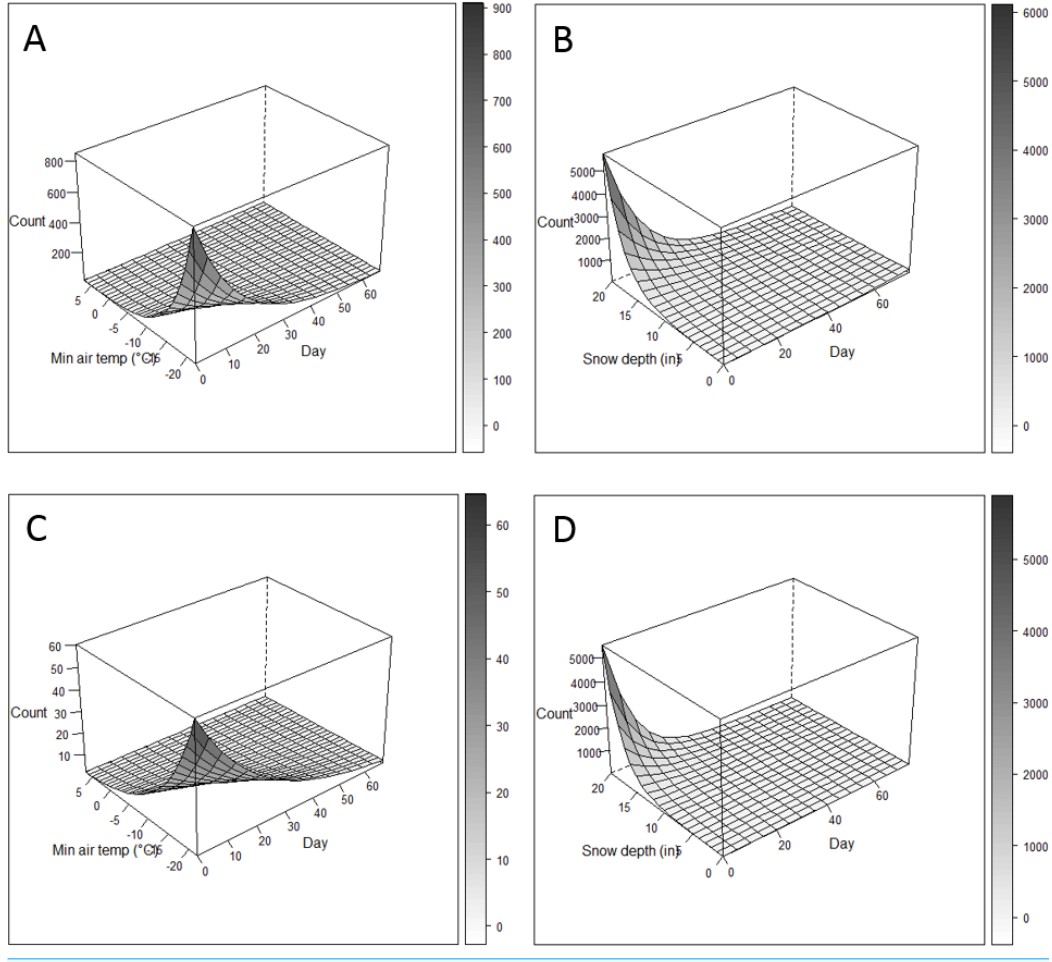

**Figure 2 Predicted use of a highway underpass based on weather conditions.** Predicted use of highway underpass during autumn migration based on most supported models of doe (A, B) and buck (C, D), over the range of weather conditions observed in 2009 (A, C) and 2010 (B, D) near Baggs, Wyoming.

average daily temperature >0 °C and no snow depth (*Garrott et al., 1987*). In the San Bernardino Mountains of southern California, mule deer exhibited partial migration in response to snow cover (*Nicholson, Bowyer & Kie, 1997*).

Installation of under and overpasses are effective in reducing deer-vehicle collisions (*Clevenger, Chruszez & Gunson, 2001*). However, under and overpasses may be cost prohibitive and thus alternative methods of migration route protection may need to be developed. In this situation, variability in the number of mule deer entering the winter range poses substantial challenges to managing mule deer-human interactions. One alternative is to use weather information to better time the use of temporary signage on roads crossed by migration routes. Temporary signage could indicate speed reduction or simply increase awareness that migration movements are more likely on days with suitable weather conditions.

In addition to providing information that could be used to more effectively manage migration route-road crossings, acquiring images of deer using underpasses and

overpasses may also increase the efficiency of monitoring efforts by state agencies. Long-term population monitoring for setting harvest regulations typically uses aerial surveys of summer grounds conducted just after the hunting season and prior to migration to wintering grounds. A concern with post-hunt aerial surveys is the potential to miss animals or have a biased sample if some classes of animals migrate prior to the survey. In these situations, underpass and overpass data may complement post-hunt aerial surveys by providing additional sampling periods during spring and fall migration. Having multiple sampling periods would enable within-year estimates of survival and recruitment as opposed to a single, annual estimate.

## CONCLUSIONS

Underpass and overpass data coupled with meteorological observations can be used to understand the relationship between weather and migration. With short-term data, we identified minimum daily temperature and snow depth as proximate cues associated with mule deer migration to winter range in Wyoming. Long-term data will provide information on migration dynamics, including whether the onset, duration, or magnitude of migration co-changes with weather. Long-term data may also reveal the role of weather in partial migration, cessation of migration or changes in route location over time.

## ACKNOWLEDGEMENTS

We thank Jerod Merkle, Stuart Pimm, and Grant Harris for comments that improved the manuscript.

### Funding

This research was supported by funding from the Wyoming Landscape Conservation Initiative, Little Snake River Conservation District, Muley Fanatics Foundation, and the University of Wyoming Student Chapter of The Wildlife Society. Avi Bar Massada provided funding for publication in PeerJ. The funders had no role in study design, data collection and analysis, decision to publish, or preparation of the manuscript.

### Grant Disclosures

The following grant information was disclosed by the authors:
Wyoming Landscape Conservation Initiative.
Little Snake River Conservation District.
Muley Fanatics Foundation.
Avi Bar Massada.
University of Wyoming Student Chapter of The Wildlife Society.

### Competing Interests

Tony W. Mong is an employee of Wyoming Game and Fish Department and Thomas Hart is an employee of Environmental Services Section, Wyoming Department of Transportation.
## Author Contributions

- Chadwick D. Rittenhouse analyzed the data, contributed reagents/materials/analysis tools, wrote the paper, prepared figures and/or tables, reviewed drafts of the paper.
- Tony W. Mong conceived and designed the experiments, performed the experiments, analyzed the data, contributed reagents/materials/analysis tools, wrote the paper, prepared figures and/or tables, reviewed drafts of the paper.
- Thomas Hart conceived and designed the experiments, performed the experiments, analyzed the data, contributed reagents/materials/analysis tools, reviewed drafts of the paper.

## Animal Ethics

The following information was supplied relating to ethical approvals (i.e., approving body and any reference numbers):

The Wyoming Game and Fish Department approved this research (Permit #791).

## Field Study Permissions

The following information was supplied relating to field study approvals (i.e., approving body and any reference numbers):

The Wyoming Game and Fish Department approved this research (Permit #791).

## Supplemental Information

Supplemental information for this article can be found online at http://dx.doi.org/10.7717/peerj.1045#supplemental-information.

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

# PeerJ

**Gagnon JW, Dodd NL, Ogren KS, Schweinsburg RE. 2011.** Factors associated with use of wildlife underpasses and the importance of long-term monitoring. *Journal of Wildlife Management* **75**:1477–1487 DOI 10.1002/jwmg.160.

**Garrott RA, White GC, Bartmann RM, Carpenter LH, Alldredge AW. 1987.** Movements of female mule deer in northwest Colorado. *Journal of Wildlife Management* **51**:634–643 DOI 10.2307/3801282.

**Ikeda T, Takahashi H, Yoshida T, Igota H, Kaji K. 2013.** Evaluation of camera trap surveys for estimation of sika deer herd composition. *Mammal Study* **38**:29–33 DOI 10.3106/041.038.0103.

**Magee L. 1990.** $R^2$ measures based on Wald and likelihood ratio joint significance tests. *American Statistician* **44**:250–253.

**Monteith KL, Bleich VC, Stephenson TR, Pierce BM, Conner MM, Klaver RW, Bowyer RT. 2011.** Timing of seasonal migration in mule deer: effects of climate, plant phenology, and life-history characteristics. *Ecosphere* **2**(**4**):Article 47 DOI 10.1890/ES10-00096.1.

**Ng SJ, Dole JW, Sauvajot RM, Riley SPD, Valone TJ. 2004.** Use of highway undercrossings by wildlife in southern California. *Biological Conservation* **115**:499–507 DOI 10.1016/S0006-3207(03)00166-6.

**Nicholson MC, Bowyer RT, Kie JG. 1997.** Habitat selection and survival of mule deer: tradeoffs associated with migration. *Journal of Mammalogy* **78**:483–504 DOI 10.2307/1382900.

**Poole KG, Mowat G. 2005.** Winter habitat relationships of deer and elk in the temperate interior mountains of British Columbia. *Wildlife Society Bulletin* **33**:1288–1302 DOI 10.2193/0091-7648(2005)33[1288:WHRODA]2.0.CO;2.

**R Core Development Team. 2012.** *R: a language and environment for statistical computing*. Vienna: R Foundation for Statistical Computing.

**Reed DF, Woodward TN, Pojar TM. 1975.** Behavioral response of mule deer to a highway underpass. *Journal of Wildlife Management* **39**:361–367 DOI 10.2307/3799915.

**Rowcliffe JM, Field J, Turvey ST, Carbone C. 2008.** Estimating animal density using camera traps without the need for individual recognition. *Journal of Applied Ecology* **45**:1228–1236 DOI 10.1111/j.1365-2664.2008.01473.x.

**Sawyer H, LeBeau C, Hart T. 2012.** Mitigating roadway impacts to migratory mule deer—a case study with underpasses and continuous fencing. *Wildlife Society Bulletin* **36**:492–498 DOI 10.1002/wsb.166.