# Peer review of "Weather conditions associated with autumn migration by mule deer in Wyoming"

_PeerJ, doi:10.7717/peerj.1045_

## Round 0.1 · original submission · Minor Revisions

Dear Chadwick:

I have been unable to get more than one reviewer so I have acted as the second one. Fortunately, I agree with Reviewer 1. This is simple, straightforward, and it's based on a very small sample size — one place and a short period of time. Now, both my reviewer and I are practical mud-on-books conservation professionals and we both know the limitations such activities pose.

Your final conclusion that one can warn motorists of increased danger from animals is a good, practical recommendation.

I want you to keep this as simple as possible. You can shorten the introduction by removing lines 12-19, shortening lines 20 to 44, and get to "our objective" (line 45) in about a paragraph.

Figure 2 doesn't work. Replace the "darker shading" with a line. Among other things, when printed you will likely lose the various shades of grey.

Figure needs larger labels to be visible.

Please let me have your changes promptly.

·

Basic reporting

This paper meets the basic reporting standard. The only comment I have is while Figure 2 is a clever plot, it's hard to interpret what is going on. And I am unsure what the reader is supposed to get out of this outside of ocular patterns (?). The paper interprets the plots but that seems rather subjective. It may be good to do away with them and stick to model results only. Further, why do the X axis scales differ between years? Maybe have a scale bar showing snow depth and temperature would help.

Experimental design

I do not think the investigation was 'rigorous', according to my definition. Reason being: 1) One camera, one site, 2 seasons of data 2) Use of explanatory variables singly or all together, and not in combinations. (so limited model sets). Why not include all possible (and valid) combinations? 3) The ratios (e.g. doe:buck, doe:faun) should have a measure of error (standard error and CI's). One way to get at this is found in "Wildlife Demography: by Skalski et al. 2005. 4) The paper should explain why analyses were combined with a day variable. I have ideas why but it would be better coming from the authors. Also, interpretation of what this day variable means in the models as it was included in top models. And, why day is in table 3 top models but not table 2? 5) Knowing which variables were correlated and what the total model included for all independent variables would be good. A table would suffice, with the threshold identified (.5, .6, .7?) 6) The models, these are generalized linear models? Right? Please be explicit here. 7) I get lost with the binary variables. Here with the sentence "We identified events as the day in which > of rain or snow fell, where value of 1 indicated event on that day;0 otherwise". What are the events cataloged for? These are separate binary variables for rain or snow on a day?

Lastly, my assumption is most deer move through the underpass one way, and don't loiter in front of cameras. So each count is likely to represent unique individuals. This said, double counting individuals may not matter in these analyses, provided this doesn't happen unique to a sex or age class. Perhaps state something to this effect, namely how "most deer move through the underpass one way, and don't loiter in front of cameras, so each count is likely to represent unique animals". Then it's explicit.

#1 above isn't going to change, outside of more effort. My hunch is that the analyses are appropriate, but given some further information as suggested above will help determine for certain.

Validity of the findings

I think the Results are valid, though as above clarifying the methods and reporting measures of error on the ratios will help.

Additional comments

This paper is short, concise and straightforward. It's not going to rock anyone's world, but it will probably help game managers in this area anticipate migration through this underpass, and probably others in similar situations. I think by clarifying the topics I brought up it will help make what was done clearer. I intend all comments to be constructive. I appreciate the opportunity to review the ms.

---

## Round 0.2 · accepted · Accept

I appreciate your detailed responses to the concerns my reviewer and I raised.